# Inequalities in the Universal Right to Health

**DOI:** 10.3390/ijerph18062844

**Published:** 2021-03-11

**Authors:** Maurizio Bonati, Gianni Tognoni, Fabio Sereni

**Affiliations:** 1Laboratory for Mother and Child Health, Public Health Department, Istituto di Ricerche Farmacologiche Mario Negri IRCCS, 20156 Milan, Italy; 2Dipartimento di Anestesia-Rianimazione e Emergenza Urgenza, Fondazione IRCCS Ca’ Granda Ospedale Maggiore Policlinico, 20122 Milan, Italy; gianni.tognoni@marionegri.it; 3Department of Pediatrics, University of Milan, 20122 Milan, Italy; Fabio.Sereni@unimi.it

**Keywords:** inequalities, human rights, health policies, epidemiological determinants, children

## Abstract

Child health inequalities violate children’s rights to optimal wellbeing. Different issues worldwide affect children’s physical and mental health as well as their development, influencing their future as adults. Inequities are avoidable inequalities. Despite improvements in the past two decades, the ambitious goals of global agendas have, for the most part, remained as expectations with regard to childhood rights, social justice, and health equity in practice. The concept of social determinants of health has become part of the common language in certain settings, but this is still too little to improve health in practice on a global scale, particularly for underprivileged subgroups of the community, as children and adolescents often are. Pediatric health professionals and their organizations are also responsible for guaranteeing children’s and adolescents’ right to health and better wellbeing, helping to reduce health inequalities.

## 1. Introduction

Inequalities are currently acknowledged as unfair and as an issue that politics must try to reduce. Inequalities that refer to either capital and income or to living conditions, however, are often considered almost as a natural consequence of modern life, where the enormous differences in needs are perceived, endured, and, in most cases, accepted. The right to education and the right to health are, today, considered by all political systems, at least theoretically, as inalienable rights, regardless of a family’s social and cultural level.

Identifying and implementing policies to correct health inequalities are complex and very demanding commitments, pursued and implemented in different ways by the various political and social realities. Inequalities in health are country-specific and are fueled by each country’s current. Their different characteristics established are associated with the national economic, social, and political profile: care systems, and forms of contrast are actually implemented. The American Pediatric Society also recently addressed an invitation to academic pediatricians to encourage them in their educational activity in specific areas of child health disparities and health equity social determinants of health [1].

The importance of socio-cultural inequalities in pediatrics is significant not only in low-income countries but also in middle-high-income countries. Is it possible to trace the main steps that have progressively transformed what seemed, in the 1970s and 1980s, an untouchable area of health law as a universal good guaranteed by national health systems into an area that sees inequality among its protagonists?

How can we understand and interpret the large amount of global and national data that document the close interaction between the health variables associated with disease management and the socioeconomic and cultural determinants of the diseases addressed, which are so heterogeneous between different countries and within the same country?

Are there current international programs of the WHO or the United Nations to refer to in order to imagine, and maybe even contribute to, the avoidance or containment of the “unwanted but inevitable effects” of inequalities?

## 2. Convention on the Rights of the Child

The CRC (Convention on the Rights of the Child) was adopted and opened for signature, ratification, and accession by the General Assembly resolution 44/25 of 20 November 1989, and it entered into force on 2 September 1990. The CRC addresses child-specific needs and rights and is the most widely ratified human rights convention. To date, 194 countries have ratified it, including every member of the United Nations except the United States. Somalia has currently not ratified it, but its government has announced that it will do so shortly. Non-member observer states (the Holy See and the State of Palestine), the Cook Islands, and Niue have adopted the CRC.

The CRC establishes the responsibility of governments, institutions, citizens, and families in ensuring that the rights of the child are respected and all actions are directed toward achieving the “best interest of the child”. The essential themes of the CRC include the right to the basic needs necessary for optimal growth and development, civil and political rights, and a right to safety and protection. Safety and protection are frequently interpreted as a freedom from forced child labor and unsafe work conditions, but they embrace a much broader focus and include safety from child maltreatment, domestic violence, and the witnessing of violence in the home and/or the larger communities. They also encompass freedom from sexual exploitation and death. With respect to health, Article 24 of the CRC obligates state parties to take appropriate measures to diminish infant and child mortality, to provide necessary health care, to combat disease and malnutrition, and to develop preventive health care.

Many country-specific studies have described specific legislative and administrative actions taken to implement the CRC that seem likely to have improved children’s rights [2,3,4]. The effects on various outcomes, however, cannot be attributed only to the CRC since changes occurred also in countries that did not adopt the CRC and in countries that adopted other core human rights treaties [5]. Most countries genuinely adopt human rights treaties and make reasonable efforts to honor their commitments under the treaties. Some countries, however, may ratify treaties strategically, with scant intentions of keeping their commitments, and may do so merely to gain some benefit, such as admission to an international organization, or to avoid criticism for continuing human rights violations [6]. Quantitative analyses of the effects of adopting human rights treaties have often failed to support their effectiveness, although the results of those analyses have varied by treaty and analysis method. Although all state parties are obliged to submit regular reports to the UN Committee on the Rights of Child on how the rights are being implemented, the outcomes of specific national interventions vary widely between countries.

## 3. Global Burden of Diseases

In the mid-1990s, with the establishment of the World Trade Organization, the world began to globalize. At the highest level of medical-scientific prominence, an acronym appeared that summarized a paradigm shift, which has since been dominant and ubiquitous: GBD (Global Burden of Diseases). The GBD is a report produced jointly by a surprising and powerful new alliance, the World Bank (which is the protagonist–promoter) and the WHO, which gives its blessing. Pathologies no longer coincide with people and populations: they are transformed into economic weights [7]. Severity, rates, distributions, projections, and infinite quantitative and qualitative epidemiological diversities coincide with a map of socioeconomic characteristics and inequalities to be taken into account in assessing the market value weights and investments that make health care one of the leading sectors of the economy. The message is explicit: medical-health interventions will be dictated and dependent on priority-sustainability assessments decided, monitored, and measured through the more or less reliable lens of GBD technology, where producing and interpreting data mean defining and controlling knowledge and decisions. Health care anticipates and experiences what has most recently been translated into a key sector of the economy: big data.

## 4. Millennium Development Goals

The depth of the paradigm shift—from the universality and equality of health-related rights to their dependence on global policies—was solemnly summed up a few years later in another acronym, MDG (Millennium Development Goals). The MDG mark the formal entry of the United Nations, which is politics tout court, in a health care system that is increasingly a mix of public and private actors who set themselves beautiful goals to change the world within the first 15 years of the Millennium [8]. Time soon documents the more than expected fragility of promises: states do not pay their contribution, private individuals (it is the beginning of the era of philanthro-capitalism, emblematized by Bill Gates) make choices based on image and interest, and the global crisis of 2008 does the rest.

The equality of rights to health, which was not only declared but also made accessible, is reaffirmed in a very authoritative (for the competence and economic representativeness of its members) document by a WHO commission that wants to recall, at least in the collective imagination, the 30 years of a document that is a symbol of equal rights, the Declaration of Alma Ata [9].

## 5. Sustainable Developments Goals

The current period we are living in, which began in 2015 and was projected to extend into 2030, is summarized in the last acronym of this path, SDGs (Sustainable Developments Goals). For the SDGs, inequality is inevitable: it is an undesirable but inexorable effect of development. The SDGs’ credibility is doubtable because they propose to eliminate poverty within 15 years, or at least to halve it [10].

In the 20 years summarized in the acronyms, inequality has not ceased to be depicted (together with its inevitable expression, inequity) in the medical-scientific literature with pathologies and environmental and cultural conditions with the most diverse names. These terms are very familiar in pediatrics: malnutrition, educational deficiencies, socioeconomic marginalization–expulsion, denied access to common goods such as water, housing, hygiene, essential diagnostic therapeutic resources, and exposure to structural and accidental pollution. A global agenda for child health and wellbeing as a blueprint for the practice of pediatrics was produced on the occasion of the 30th anniversary of the UN Convention on the Rights of the Child [11].

## 6. Social Determinants of Health

The objective summarized in the commission’s acronym, SDH (Social Determinants of Health), is clearly in contrast with the dominant political evolution [12] and was produced by a mixture of ideas. These ideas originated from both a culture that sees health as an inseparable expression of universal and equal human rights, and policies, supported also by an invasion of “scientific” publications, that aim to make people accept a type of health that corresponds to an “insurance coverage” that policyholders promise will be universal: Universal Health Coverage (UHC). The real world is different, however, as indigenous health and inequalities between indigenous and non-indigenous populations internationally, and within the same country, show us [13]. Indigenous peoples’ health is intimately linked to the social and political environment in which they live. Thus, the determinants of the health of indigenous individuals are linked to the determinants of community and ecosystem health—a holistic notion that is key to social determinants of indigenous health and is linked fundamentally to “indigenous” identity [14]. This example points out the substantial limitations of existing evidence linking social determinants of specific population groups to specific health outcomes. Greater efforts should be made to address inequalities also in indigenous children [15].

## 7. Health Inequalities

Health inequalities are the unjust and avoidable differences in people’s health across the population and between specific population groups. The impact of inequality is stronger and more certain than that of poverty. The unequal live in proximity but belong to different worlds. They can be citizens who cross each other on the same roads but have absolutely incomparable living conditions and needs.

The synonyms of inequality mentioned above are a pandemic that has no borders. In large cities, the measures that document that life expectancy (not just quality) decreases, in terms of years, in proportion to the distance by subway from “residential” areas, are now classic.

One of the important determinants of health inequalities within society is the availability and nature of employment. Employment is linked to the fundamental causes of health inequality—the unequal distribution of income, wealth, and power. Increasing the quality and quantity of work can help reduce health inequalities [16]. Similarly, there is a strong link between higher levels of education and higher self-rated health rates, lower morbidity, and better access to health care. Thus, poorly educated citizens are more likely to be at risk of poverty or social exclusion than those with tertiary-level education [17]. Increasing the societal level and the education-based social mobility will increase societal equity, also in health.

No medical training text dedicates significant chapters to the transversality of inequality. Data on the impact of economic inequality on health indicators, however, occupy an important space in the texts of mainstream economists.

The questions are clear and simple: Is all this avoidable? Can health care do something to avoid, control, “cure”, or rehabilitate from inequality, if it is, at the same time, an expression of it, for reasons unrelated to medical expertise?

## 8. From Present to Future

In the recent literature, the relevant papers are a few dozen and rarely of quality [18]. The GBD’s descriptive logic is absolutely dominant. The various causes of inequality are presented and analyzed, from malnutrition to mortality, and from diarrhea to pollution; the evidence described is almost irritating because of the precision with which one expects to provide the explanations, which all then simply end with recommendations and resolutions [19].

Only one article focuses on the decisive element of avoidability: “32% of children under 5 in low- and middle-income countries lived in districts that had attained rates of 25 or fewer child deaths per 1000 live births by 2017, and that 58% of child deaths between 2000 and 2017 in these countries could have been averted in the absence of geographical inequality” [20].

Furthermore, there are two examples that formally explore the possible causes of a higher child mortality rate in the UK compared to Sweden [21,22]. Child mortality rates are significantly higher in the UK than in Sweden. The majority of these infections are treatable, so interventions should focus on service delivery and access [21]. Socioeconomic factors also contribute to these differences through associations with adverse birth characteristics and increased mortality after 1 month of age. Actions should therefore focus on improving the health of women before and during pregnancy and in reducing socioeconomic disadvantage [22]. When children have to cope with social, health, and family adversities, however, they are also exposed to a considerably higher risk of mortality in early adulthood than other children, as reported in a Danish population study [23]. Despite the considerable work conducted, for example, the efforts made in the UK to publicize and tackle inequalities in health, such as the use of child welfare benefits and early intervention initiatives such as Sure Start [24], efforts to reduce child health inequalities need perseverance over time and the investment of further resources. Interventions to contrast these adversities should therefore begin early and continue throughout life, from the prenatal period to old age. These interventions should be present in all countries, even those with high resources, since the determinants of inequalities are ubiquitous and perennial.

It is surprising that some areas of medicine, such as mental health in the developmental age, are still neglected and are characterized by substantial gaps in resources for patients [25]. Despite the repeatedly declared intentions, the initiatives undertaken for the improvement of child mental health care services are partial (consisting mainly of projects) and local, focusing on individual disorders and not on national strategic interventions [26]. The lack of acknowledgement of common standards of care and quality indicators, or of possible benchmarks for local services, not only represents a limit in guaranteeing the “essentiality” and appropriateness of care but also leads to the maintenance of chronic inequalities, between and within countries. This perpetuation of chronic inequalities occurs in the questions, but even more in the answers, related to quality and appropriateness. Mental health equity for children and adolescents is the foundation for the future. The difficulties with, or lack of access to, mental health services increase social inequalities in children’s mental health. Nevertheless, social determinants are associated with inequalities in mental health independently of access to services. Multimodal interventions with potential to prevent and reduce health inequalities therefore remain a public health priority worldwide. Unfortunately, the prediction is that inequity in child health will negatively impact society for decades to come [27].

The dissociation between the actual invasion of inequality in the literature and the absence of research interest and results aimed at actual populations could not have been more impressive.

In this regard, a small, but significant, core of recent British articles (not dedicated to, but which also include, pediatrics) is worth reading [20,28,29,30,31,32].

## 9. Being Born and Growing Up in Inequality

Health is unequal between and within nations, and this has long been well known to policy makers, as has the fact that inequalities affect multiple areas and contexts of people’s lives, i.e., life expectancy, disease, access to education, employment, income, housing conditions, and healthy habits. The policies put in place to combat avoidable health inequalities, and the appropriate prevention interventions sustained over time, however, have not been implemented adequately, especially for fragile populations such as the pediatric one.

Until the early 2000s, health inequalities were measured and compared between nations mainly in terms of mortality and, for the pediatric population, in terms of infant mortality in the first 5 years of life. The mortality rate was the independent indicator associated with per capita income, the Gini coefficient, and other dependent indicators also in the construction of indices such as the Human Development Index [33]. The use of macro indicators is limited due to the genericity (low precision and specificity) of the estimates of the indicators at the national level and, consequently, the genericity of both temporal and national comparisons. This is an approximation of reality that is further questioned with the accumulation of evidence on social inequalities in health between, and within, countries and on the causal factors likely to induce them [32].

Children born in disadvantaged socioeconomic conditions suffer from poorer wellbeing throughout their lives, in all societies around the world [34]. Evidence of the association of social determinants of health (e.g., relative poverty and income inequality) with the wellbeing of children, and with the public health of an entire population, however, exists. Politics nevertheless tends to focus on interventions aimed at mitigating the effects, rather than on removing or countering the causes. Pediatricians, aware of the impact of the social determinants of health on children’s growth, must therefore contribute to addressing inequalities in health in the developmental age also for social justice [35].

Concerning social determinants, even in the pediatric area, the territorial contexts within individual nations have become increasingly variable and dependent on the state of wellbeing of the inhabitants. Despite the substantial reduction in infant mortality in all nations since 2000, the extremes, represented by Cuba with 5.1 deaths in the first 5 years of life per 1000 children and the Central African Republic with 123.9 deaths, indicate that improvements are still necessary [20]. The risk of an infant dying is associated with geographic context, conditions of care, and prenatal determinants. Everywhere, the risk is greatest in the perinatal and neonatal periods.

In addition to social determinants, there is also the social gradient in child health within countries [36]. However, there is some limited and inconsistent evidence that health is worse at all points of the social gradient in more unequal countries [37].

## 10. Perspectives

A few of objectives of key initiatives aimed at reducing child health inequalities have been achieved, such as initiatives taken at the level of the Commission on Social Determinants of Health, the WHO Regional Office, and the European Commission. Some reports have documented these efforts and findings [38,39]. International reports have already referred to the problem of health inequalities in their titles [40,41,42,43]. Child health inequalities, however, persist also in countries and regions where contrast strategies have been applied and implemented. 

Is it possible, and how, to make inequality one of the central axes of pediatric practice and research, not only in cultural terms but also in the interventions to counter it?

Clinicians, pediatricians in particular, have to make an effort for the health and wellbeing of all children [44].

Finding a way to make inequality a component of teaching is a challenge that needs to be addressed immediately, not by setting up an academic course in epidemiology or public health, but as an essential part of a training course aimed both at combating inequalities and their effects on health, and at promoting the dignity of life of pediatric populations.

An epidemiological map of the impact of inequalities on the pediatric population in the different care and life contexts should be a regularly updated tool that guides health planning. This tool should be commonly and largely acknowledged, not only between different professions (medical, sociological, economic, and legal) but also between different community, administrative, and political contexts. The many data available can, and must, shift from their descriptive role to being an instrument of participation and shared evaluation.

It is time for new global child and adolescent research, particularly research in mental health [45]. There is no doubt that the research strategies, which are essential and priority, must also favor the creation of collaborative networks in this field. These research strategies should be representative of the diversity of care and life contexts and should keep in mind that the risk of inequality is inversely related to the presence of experimental interventions to reduce, if not avoid, it.

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
