# Peer review of "Inequalities in the Universal Right to Health"

_ijerph, 2021, doi:10.3390/ijerph18062844_

Round 1

Reviewer 1 Report

Review report

  1. Inequalities  in health are country specific and they are fuelled by  the government of the day in each country. This aspect was not well articulated in the manuscript.
  2. In section 6 social determinants of health, the indigenous knowledge system of health followed by many African population particularly those in the rural area if highlighted could be of interest to some readers.
  3. Health inequalities of section 7 goes hand  with the academic level of the society and the employment status. I suggest the author briefly highlights the relationship in the three areas and show how they could be shaped to improve the health system
  4. A few of objective of the key initiatives at reducing child health have been achievable according to the authors. One wonders the practicalities of these statement if it is back by  data that is country specific.

Author Response

Thank you for the suggestions that contribute in improving the text.

   Inequalities  in health are country specific and they are fuelled by  the government of the day in each country. This aspect was not well articulated in the manuscript.

see lines 36-40 for what added.

    In section 6 social determinants of health, the indigenous knowledge system of health followed by many African population particularly those in the rural area if highlighted could be of interest to some readers.

see lines 157-167 for what added. Three pertinent references were also added.

    Health inequalities of section 7 goes hand  with the academic level of the society and the employment status. I suggest the author briefly highlights the relationship in the three areas and show how they could be shaped to improve the health system

see lines 179-188 fro what added. Two pertinent references were also added.

    A few of objective of the key initiatives at reducing child health have been achievable according to the authors. One wonders the practicalities of these statement if it is back by  data that is country specific.

see new lines 36-40 that include the point.

Reviewer 2 Report

This commentary addresses very central and important issues related to health, namely health as a universal good that should be guaranteed by law, the understanding of complex sets of variables linked to disease management, and the international as well as internal effects of inequalities on children’s universal right to health.

The argument is as follows: The international legal framework (UNCRC) is reminded, and the emergence of a paradigm shift in medical-scientific approach is underlined with the GBD (Global Burden of Diseases) putting priority-sustainability assessments to the fore. Knowledge and decisions in the medical sector are more and more shaped by big data, and translated in Millenium Development Goals (MDG). The growing influence of the UN on health is marked by the acceptance of the inevitability of inequalities, putting SDG’s credibility in doubt (elimination of poverty).

The authors suggest that health inequalities, by contrast, are avoidable. The impact of inequality is stronger than that of poverty. Hence, they deplore that “No medical training text dedicates significant chapters to the transversality of inequality”, and substantiate this statement in their literature review (section 8). Further, they deplore that health policies have not been adequately put in place, centred on symptoms and not causes, and questioning the indicator of infant mortality rate.

The models have to include inequality among the social determinants of health. This begins with the teaching cursus of pediatricians, that has to include “an epidemiological map of the impact of inequalities on the pediatric population in the different care and life contexts. The authors claim for less description and more participation and shared evaluation. They stress the need for more collaborative child and adolescent research as this might be inversely proportional to the risk of inequality.

The commentary is very well structured and the arguments are all convincing. It bases on a wide knowledge and literature review of the evolution of Pediatrics and Health in general and makes relevant connections with human rights instruments (UNCRC), and necessary adaptations of UN instruments and research paradigms. The article is clear and reads well. The only word that is problematic is “consociative” (line 27). Its significance is not clear, although one might guess it might be an “Italianism” for the word “associative”. This word should be either corrected or specified.

Author Response

Thank you very much for your considerations of our work.

The only word that is problematic is “consociative” (line 27). Its significance is not clear, although one might guess it might be an “Italianism” for the word “associative”. This word should be either corrected or specified.

see new lines 29-30. We canceled "consociative" word, and we attempted to explain theconcept.